# APE-Bench: Evaluating Automated Proof Engineering for Formal Math Libraries

Huajian Xin [1 2]    Zheng Yuan [1]    Jacques D. Fleuriot [2]    Wenda Li [2]

## Abstract

While frontier formal mathematics systems now routinely develop repository-scale proof engineering artifacts requiring multi-file coordination and semantic correctness beyond compilation, existing evaluation benchmarks remain focused on isolated theorem proving. We introduce Automated Proof Engineering (APE), the first systematic framework for evaluating repository-scale proof engineering through dual verification that validates both syntactic compilation and semantic requirement satisfaction in pinned library environments. We present a complete infrastructure comprising APE-Bench, which automatically extracts proof engineering tasks from real library commit histories, and APE-Harness, a unified execution framework based on task contract abstraction. This contract-based design enables standardized evaluation across diverse formal mathematics tasks and fair systematic comparison of different agent implementations (including our APE-Agent reference scaffold alongside Claude Code and Codex CLI) on identical task specifications. We demonstrate the framework's effectiveness through comprehensive evaluation. All code, benchmarks, and infrastructure are released as open-source at https://github.com/xinhjBrant/APE-Bench.

## 1. Introduction

Research in LLM-based formal mathematics has long centered on theorem proving: given a formal statement, synthesise a proof term that type-checks. Yet the practice of formal mathematics development has evolved far beyond isolated proving. Production libraries like Mathlib now span millions of lines (mathlib Community, 2020; Baanen et al., 2026), and even AI systems focused on theorem proving require repository-scale work: proving a difficult theorem involves searching codebases for lemmas, coordinating auxiliary results across declarations and files, and contributing to libraries (Math, Inc., 2025; Sothanaphan, 2026; Chen & et al., 2025). The interactive theorem proving community terms this broader discipline *proof engineering* (Ringer et al., 2019): software engineering for proofs. As formal developments scale to millions of lines, they face challenges of library evolution, cross-file coordination, and maintenance under change—challenges that isolated theorem proving does not address. Yet existing benchmarks such as miniF2F (Zheng et al., 2022) and FATE (Jiang et al., 2025) remain focused on isolated theorem proving, evaluating proof synthesis for given statements rather than the repository-scale engineering activities that characterize real formal mathematics development.

We introduce **Automated Proof Engineering (APE)** as a task formulation that operationalizes repository-scale proof engineering for systematic evaluation. Each APE task specifies a pinned repository and toolchain environment, a natural-language instruction describing the objective, and dual verification: syntactic correctness via proof-checker compilation and semantic correctness confirming requirement satisfaction within declared scope. This mirrors how SWE-bench (Jimenez et al., 2023) requires both code compilation and test suite passage for software engineering tasks.

This paper provides a complete infrastructure for evaluating repository-scale proof engineering through three integrated components (Figure 1). **APE-Bench** instantiates the APE formulation through an automated pipeline that mines real Mathlib commit histories (Figure 1, top): each task extracts a file-level modification from actual maintenance commits, distilling developer intent into natural-language instructions while preserving original repository and toolchain states. The pipeline enables continual regeneration as Mathlib evolves. **APE-Harness** provides an execution and evaluation infrastructure built on *task contracts* that separate task specification from execution strategy across diverse formal mathematics activities (Figure 1, bottom). A task contract specifies environment bindings, objectives, boundaries, and verification protocols as declarative specifications.

[1]ByteDance Seed [2]School of Informatics, University of Edinburgh, Edinburgh, UK. Correspondence to: Huajian Xin <h.xin-3@sms.ed.ac.uk>.

*Proceedings of the 43rd International Conference on Machine Learning*, Seoul, South Korea. PMLR 306, 2026. Copyright 2026 by the author(s).

The infrastructure provides execute services for compilation verification, retrieve services for semantic search, and runtime orchestration enforcing workspace isolation and verification protocols.

The key design insight underlying APE-Harness is the task contract abstraction that separates task specification from execution strategy. Compared to progress-tracking patterns (e.g., Claude Code (Anthropic, 2025)) that specify tasks through only natural-language instructions and tool declarations, APE-Harness adds programmatic enforcement (workspace initialization, access control, verification protocols), while remaining execution-agnostic unlike execution-encapsulation patterns (e.g., LangGraph (LangChain, 2024)) that embed complete implementation logic. This design enables three critical capabilities. (a) First, unified evaluation across diverse task types: the contract abstraction enables traditional theorem proving, proof engineering with semantic validation, and other formal mathematics evaluation tasks to execute through the same infrastructure components—shared tool services and runtime orchestration. (b) Second, beyond evaluation to functional workflows: instruction synthesis for benchmark construction and library annotation for retrieval are formulated as task contracts, demonstrating infrastructure generality beyond evaluation. (c) Third, scaffold interchangeability and composability: by specifying requirements without prescribing execution strategies, contracts enable different agent scaffolds (APE-Agent, Claude Code, Codex CLI) to execute identical tasks while accessing shared verification services. Contracts also compose through nested execution: proof engineering tasks invoke judgment tasks as semantic validation sub-components, with semantic validation itself executing as task contracts through APE-Harness. This self-hosted design validates that infrastructure components can be executed through the infrastructure itself.

We evaluate frontier models (GPT-5.2, Gemini 3 Pro, Gemini 3 Flash) on APE-Bench using the APE-Harness infrastructure. For this evaluation, we extract 100 tasks from 67 Mathlib commits dated 2026-01-06 through 2026-01-12, ensuring zero training data contamination. Since each task must execute in its original commit environment, this design requires supporting compilation and retrieval across 67 library versions. We address this through a centralized content deduplication system that shares identical files and declarations across versions, achieving order-of-magnitude resource savings.

Beyond the core APE-Bench evaluation, we validate the infrastructure through multiple dimensions. First, we evaluate models on traditional theorem proving benchmarks (miniF2F, miniCTX) to demonstrate the infrastructure's generality beyond proof engineering. Second, we validate semantic judge reliability and compare scaffold im-

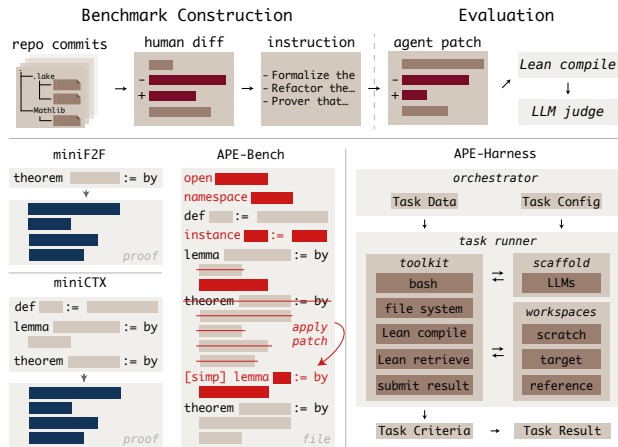

*Figure 1.* **APE framework overview. Top:** Benchmark construction and evaluation pipeline. **Bottom:** Evaluation paradigm comparison (miniF2F/miniCTX, APE-Bench) and APE-Harness infrastructure architecture (task contracts, scaffolds, toolkits, workspaces).

plementations (APE-Agent, Claude Code, and Codex CLI) to demonstrate contract abstraction effectiveness. Third, we demonstrate multi-version infrastructure efficiency through content deduplication, enabling practical evaluation across 67 library versions.

**Contributions.** We provide the first complete infrastructure for repository-scale proof engineering evaluation:

1. **APE Formulation and Benchmark.** We formalize Automated Proof Engineering with pinned repository-/toolchain environments, natural-language instructions, and dual verification. We provide an automated pipeline (APE-Bench) that mines proof engineering tasks from Mathlib commit histories, constructing 100 tasks from 67 recent commits ensuring zero training data contamination.

2. **LLM Judge Validation Benchmark.** We construct 64 expert-annotated agent solutions with three-dimensional quality ratings (semantic correctness, requirement alignment, scope control) to validate that our LLM-as-Judge pipeline aligns with human expert judgment, ensuring reliable semantic validation in APE-Bench.

3. **APE-Harness: Task Contract Evaluation Infrastructure.** We provide execution infrastructure based on task contracts that abstract task specification from execution strategy, enabling unified evaluation across diverse tasks, self-hosted functional workflows, and scaffold interchangeability.

4. **APE-Agent: Scaffold for Research.** We provide a ReAct-style scaffold featuring file system tools with integrated verification. This white-box implementation

facilitates research and extension, while enabling comparison with mature code agents (Claude Code, Codex CLI).

5. **Multi-Version Execution and Retrieval Services.** We implement execute and retrieve services supporting compilation verification and semantic search across multiple library versions, achieving order-of-magnitude resource savings through content deduplication.

## 2. Related Work

**Formal mathematics benchmarks.** Established benchmarks for formal mathematics evaluation focus on proof synthesis given formal statements. miniF2F (Zheng et al., 2022) and FATE (Jiang et al., 2025) provide formal statements and require agents to synthesise proof terms that type-check. miniCTX (Hu et al., 2024) extends evaluation with repository-level retrieval to support proof synthesis in long context, yet preserves this task structure.

**Proof engineering tools.** The interactive theorem proving community has developed tools to address proof maintenance challenges at scale. iCoq (Celik et al., 2017) automates proof repair under definition changes through regression proving; REPLICA (Ringer et al., 2021) enables proof reuse across library evolution through program transformations. However, these tools remain largely heuristic and require significant manual effort for complex repository-scale changes.

**Proof engineering in practice.** Recent AI systems for theorem proving exemplify the reality of proof engineering at scale: while their objective remains proving theorems, their actual work constitutes repository-scale engineering. Math Inc's Gauss (Math, Inc., 2025), Harmonic's Aristotle (Sothanaphan, 2026), and Seed-Prover (Chen & et al., 2025) coordinate auxiliary results across declarations and files, manage developments through structured blueprints, and contribute to production libraries like Mathlib (mathlib Community, 2020). Yet systematic evaluation frameworks for these repository-scale capabilities remain absent.

**Repository-scale evaluation paradigm.** Software engineering has addressed analogous challenges through repository-scale evaluation. SWE-bench (Jimenez et al., 2023) extracts tasks from real commits with natural-language descriptions and verifies both compilation and test passage. Code agents have proven successful in this paradigm (Yang et al., 2024; Anthropic, 2025). APE adapts this paradigm to formal mathematics, replacing test suites with dual verification.

## 3. Automated Proof Engineering

Automated Proof Engineering (APE) formulates repository-scale proof development as instruction-driven code modification (Figure 1, bottom). Each task specifies a natural-language instruction, a pinned environment (repository commit and toolchain), and dual verification requiring both compilation and semantic validation. Figure 2 illustrates a task requiring proof automation and filter lemmas for square root limits.

---

**APE Task Example: Square Root Limits and Automation**

**Repository:** `mathlib4@622f41ab`

**Toolchain:** `leanprover/lean4:v4.27.0-rc1`

**File:** `Mathlib/Data/Real/Sqrt.lean` (477 lines, blocked)

**Instruction:**
Add limits at infinity for Real square root and automate continuity proof:

1. Automate the proof of `continuous_sqrt` and register it with `fun_prop`.

2. Prove that the image of the `atTop` filter under square root is `atTop`.

3. Prove that the preimage (comap) of the `atTop` filter under square root is `atTop`.

4. Establish that the square root function tends to infinity at infinity.

**Key modifications required (13 lines changed):**

```
@@ -122,8 +122,11 @@
-@[continuity]
-theorem continuous_sqrt : Continuous (√· : ℝ →
    ℝ) := by
-   unfold sqrt
-   exact NNReal.continuous_coe.comp <|
-     NNReal.continuous_sqrt.comp
     continuous_real_toNNReal
+@[continuity, fun_prop]
+theorem continuous_sqrt : Continuous (√· : ℝ →
    ℝ) := by unfold sqrt; fun_prop
+
+@[simp]
+lemma map_sqrt_atTop : map (√·) atTop = atTop
    := by
+   unfold sqrt; change map (toReal o sqrt o
    toNNReal) atTop = atTop; simp [← map_map]
+
+@[simp] lemma comap_sqrt_atTop : comap (√·)
    atTop = atTop := by [...]
+lemma tendsto_sqrt_atTop : Tendsto (√·) atTop
    atTop := map_sqrt_atTop.le
```

*Figure 2.* **APE task example.** This task requires automating `continuous_sqrt` with `fun_prop` and proving three filter properties at infinity.

**Task components.** An APE task comprises three integrated components. The *instruction* describes modification objectives admitting multiple implementations. The *environ-*

*ment* pins repository commit and toolchain, with the target file blocked while preserving library access. *Verification* requires both compilation (modifications type-check) and semantic validation (objectives met).

**Semantic validation dimensions.** Compilation alone proves insufficient for proof engineering evaluation. Semantic validation addresses three distinct failure modes through judgment-based assessment (Section 4). *Requirement alignment* verifies instruction objectives are fully satisfied—type-checking is necessary but not sufficient for correctness. For example, in Figure 2, solutions must use `fun_prop` for automation, register the attribute, and include all four required lemmas. *Scope control* distinguishes focused engineering from undisciplined refactoring—solutions must achieve stated goals without gratuitous changes, such as automating `continuous_sqrt` without refactoring unrelated theorems. *Semantic correctness* validates logical validity beyond compilation—mathematical properties must hold, not merely type-check, ensuring filter lemmas establish correct mathematical properties rather than arbitrary type-correct formulations.

**Relation to software engineering evaluation.** APE adapts SWE-bench's (Jimenez et al., 2023) evaluation paradigm to proof engineering, replacing unit tests with judgment-based validation since proof tasks lack pre-existing test coverage (Table 1).

| Task Category | Software Engineering | Proof Engineering |
|---|---|---|
| Input | GitHub issue + repo | Instruction + file + repo |
| Syntactic Check | Compilation / interpretation | Lean compilation |
| Semantic Check | Unit test passage | LLM-as-judge |

*Table 1.* Comparison of software engineering and proof engineering evaluation.

## 4. APE-Harness: Evaluation Infrastructure

APE-Harness provides execution infrastructure for repository-scale proof engineering tasks (Section 3, Figure 1 bottom). Built on task contract abstraction separating specification from execution strategy, the infrastructure enables unified evaluation across diverse task types, extension to functional workflows (Section 5 demonstrates), and scaffold interchangeability (Section 6 validates). This section details the contract abstraction (§4.1), infrastructure services (§4.2), and scaffold implementations (§4.3).

### 4.1. Task Contract Abstraction

**Design rationale.** Systematic evaluation of repository-scale proof engineering requires executing diverse task types across heterogeneous environments. Tasks range from theorem proving to proof engineering with semantic validation to judgment assessment, each pinned to specific repository commits and toolchain versions. Agents working in these varied environments cannot rely on memorized library APIs from training data; instead, they must discover and adapt to available components at runtime. This adaptation emerges from infrastructure design: file operations (varying by scaffold) enable exploring repository context, execute services verify correctness against pinned installations, and retrieve services discover available declarations. Supporting this diversity while maintaining evaluation consistency requires abstracting task specification from execution strategy—the foundation of contract-based infrastructure design.

**Contract specification.** A task contract comprises four declarative components: (i) *Environment bindings* specify repository commit hash and toolchain version, ensuring reproducibility; (ii) *Objectives* define agent goals—theorem statements for proving, natural-language instructions for engineering, assessment criteria for judgment; (iii) *Boundaries* declare access constraints through read-only and blocked path patterns, preventing solution leakage while preserving compilation context; (iv) *Verification protocols* define success criteria, from compilation checking to dual verification combining syntactic and semantic validation. Contracts specify requirements without prescribing strategies, enabling task-agnostic infrastructure components.

**Task abstraction design comparison.** Task abstractions in agent frameworks embody different architectural trade-offs. Table 2 provides an overview:

| Claude Code | Natural-language instructions, tool specifications. |
|---|---|
| **APE-Harness** | Natural-language instructions, tool configurations, programmatic workspace initialization, environment version bindings, access control, verification protocols. |
| **LangGraph** | Hardcoded execution flows directly in Python function. |

*Table 2.* Task abstraction components overview.

- *Progress-tracking pattern* (e.g., Claude Code (Anthropic, 2025)): Tasks specify goals through natural-language descriptions and tool declarations, relying on prompts for verification guidance. APE-Harness augments this with programmatic enforcement (workspace initialization, access control, explicit verification protocols) while remaining execution-agnostic.

- *Execution-encapsulation pattern* (e.g., Lang-Graph (LangChain, 2024)): Tasks embed complete execution logic within decorated functions to enable deterministic replay. APE-Harness separates specification from execution: contracts declare requirements and verification without prescribing implementation strategies, enabling scaffold interchangeability.

## 4.2. Infrastructure Services

**Multi-version execution.** Execute services provide compilation verification by compiling agent code from *scratch* workspaces against immutable *target* library installations at contract-specified commits, returning structured diagnostics. Supporting verification across diverse repository versions requires efficient management of compiled library artifacts. The infrastructure organizes these through content-addressable storage: library files are indexed by cryptographic hash rather than version identifier. When multiple library versions contain identical files, they share compiled artifacts through common hashes. Lightweight version manifests map each commit to its constituent content hashes without duplicating artifacts. This design transforms linear version scaling into logarithmic growth, achieving order-of-magnitude resource savings.

**Version-scoped retrieval.** Retrieve services enable discovering library components through semantic search scoped to target versions. Agents query through three modes: exact name matching for known identifiers, semantic search using natural-language descriptions, and keyword-based topical search. All results filter to declarations existing at the contract-specified commit, ensuring agents discover only available components. Semantic search requires annotating declarations with natural-language descriptions. Annotation generation is formulated as task contracts: objectives specify generating descriptions for library declarations, environment bindings pin target library versions, boundaries restrict to read-only file tools, and verification requires submitting annotations through structured tool interface. This demonstrates extension to functional workflows beyond evaluation, with instruction synthesis (Section 5) similarly formulated.

**Workspace isolation.** Contract boundaries are enforced through physical file system separation. *Scratch* workspaces provide writable directories where agents explore and accumulate artifacts. *Target* workspaces contain immutable library installations—agent code compiles against these paths but cannot modify them. *Reference* workspaces offer read-only access to related dependencies. OS-level permissions prevent bypassing contract-specified boundaries.

## 4.3. Scaffold Implementations

**APE-Agent scaffold.** We provide a ReAct-style scaffold (APE-Agent) with file system tools featuring integrated verification: edit operations atomically return compilation results rather than requiring separate verification calls. When agents modify files, they immediately receive diagnostics indicating whether changes compile successfully. This tight perception-action loop enables rapid iterative refinement in persistent *scratch* workspaces.

**Scaffold interchangeability.** Task contracts enable scaffold interchangeability by defining tasks as declarative specifications independent of execution strategies. Different scaffolds (APE-Agent, Claude Code, Codex CLI) execute identical tasks specified by the same contracts—the same environment bindings, objectives, boundaries, and verification protocols. Each scaffold employs its own tools and implementation approach, but all must satisfy identical contract requirements. This enables principled comparison where performance differences reflect scaffold effectiveness rather than task definition variations.

## 4.4. Task Contract Instantiations

Table 3 summarizes task contract instantiations. Three properties distinguish contract-based design: *Unified interface.* All tasks execute through identical infrastructure, differing only in contract specifications. *Composability.* Contracts compose through nested execution: proof engineering invokes judgment as validation sub-tasks. *Self-hosted execution.* Instruction synthesis and library annotation execute as agent workflows through APE-Harness, enabling tool-based exploration rather than single-shot generation. Infrastructure extends beyond evaluation to functional workflows.

| Task Type | Description |
|---|---|
| Theorem Proving | Synthesises proofs for formal statements (miniF2F, miniCTX benchmarks) |
| Proof Engineering | Repository-scale code modification guided by natural-language instructions (APE-Bench evaluation) |
| Judgment | Assesses semantic correctness, requirement alignment, and scope control; invoked by proof engineering for semantic validation |
| Instruction Synthesis | Generates task descriptions from commit diffs for benchmark construction (Section 5) |
| Library Annotation | Generates semantic descriptions for declaration retrieval (§4.2) |

*Table 3.* Task contract instantiations across evaluation and functional workflows.

# 5. APE-Bench: Automated Benchmark Construction

APE-Bench is an automated pipeline for extracting proof engineering tasks from commit histories (Figure 1, top). The pipeline operates parametrically over repository and commit range, enabling benchmark instantiation for different scenarios and continual regeneration as libraries evolve. Section 6 demonstrates instantiation on recent Mathlib commits ensuring training data contamination barriers.

## 5.1. Pipeline Stages

The pipeline operates in three stages: commit filtering extracts candidate edits from repository histories, instruction synthesis generates natural-language task descriptions from diffs, and validation ensures synthesised instructions are solvable.

**Commit filtering.** Given a repository and date range, the pipeline extracts file-level modifications constituting coherent engineering operations. Filtering criteria target substantive changes: diffs spanning 10–300 lines with at least 10 non-blank added lines, affecting mathematical content, and type-checking in post-edit environments. Multi-file commits decompose into file-level tasks, and scattered edits lacking coherent intent are filtered.

**Instruction synthesis.** Formulated as task contracts and executed through APE-Harness, instruction synthesis generates natural-language task descriptions from commit diffs. Agents explore commit contexts through file tools—examining pre/post-edit files, searching related declarations—then submit structured instructions comprising title, objectives, and implementation guidance. The instruction format targets actionability (clear objectives), generalizability (describing intent admitting alternative solutions), and groundedness (accurate reflection of changes). For example, a commit automating a manual proof yields "automate Theorem X using newly available Lemma Y" rather than "modify lines 23–72 as shown"—the former enables evaluation through intent satisfaction while the latter reduces to diff matching. Success requires valid submission format and optional validation that the original patch satisfies the generated instruction under dual verification (Section 3).

## 5.2. Pipeline Features

The pipeline exhibits three key properties distinguishing it from manual benchmark construction:

- *Self-hosted through infrastructure.* Instruction synthesis executes as task contracts through APE-Harness (Section 4), demonstrating infrastructure extension to functional workflows.

- *Automated extraction and validation.* The pipeline automates task generation from commit histories without manual annotation. Agents explore repository context through file tools, avoiding reliance on memorized knowledge. Automated validation ensures every task is solvable, enabling large-scale generation.

- *Continual regeneration.* Pipeline parameterization over commit ranges enables contamination control (evaluators choose commits after model training cutoffs) and continual evolution (newer commits expand task coverage as libraries grow). Section 6 demonstrates instantiation on recent commits avoiding contamination.

# 6. Evaluation

We evaluate APE through multiple dimensions: (1) model performance and efficiency analysis establishing cost-performance tradeoffs across budget constraints; (2) semantic judge reliability validation; (3) scaffold interchangeability demonstration; (4) tool configuration ablation; (5) multi-version infrastructure efficiency validation.

## 6.1. Experimental Setup

**Benchmark instantiation.** We instantiate APE-Bench on Mathlib commits from 2026-01-06 through 2026-01-12, producing 100 tasks spanning 67 commits with modification sizes ranging from 10–284 diff lines (median 30). We additionally evaluate on miniCTX (334 tasks) and miniF2F (244 tasks) to validate infrastructure generality across task paradigms.

**Models and configurations.** We evaluate three frontier models (GPT-5.2, Gemini 3 Pro, Gemini 3 Flash) across three scaffolds (APE-Agent, Claude Code, Codex CLI). Each evaluation run uses 100-turn limit with $3 budget per task and retrieval enabled (top-10 declarations). We track cumulative cost and dialogue turns at each task completion or failure, enabling efficiency curve analysis. Primary results use APE-Agent scaffold; §6.5 validates scaffold interchangeability. Agents have no internet access during evaluation and interact exclusively with APE-Harness infrastructure services (file operations, compilation, declaration search) within isolated workspaces.

**Verification methodology.** All tasks undergo dual verification: compilation against pinned repository/toolchain environments ensures syntactic correctness, while semantic validation through LLM judges confirms requirement satisfaction and scope control. Judges execute as nested task contracts (§4.4) with majority voting across 3 judges (GPT-5 Mini) producing final decisions. §6.4 validates judge accuracy.

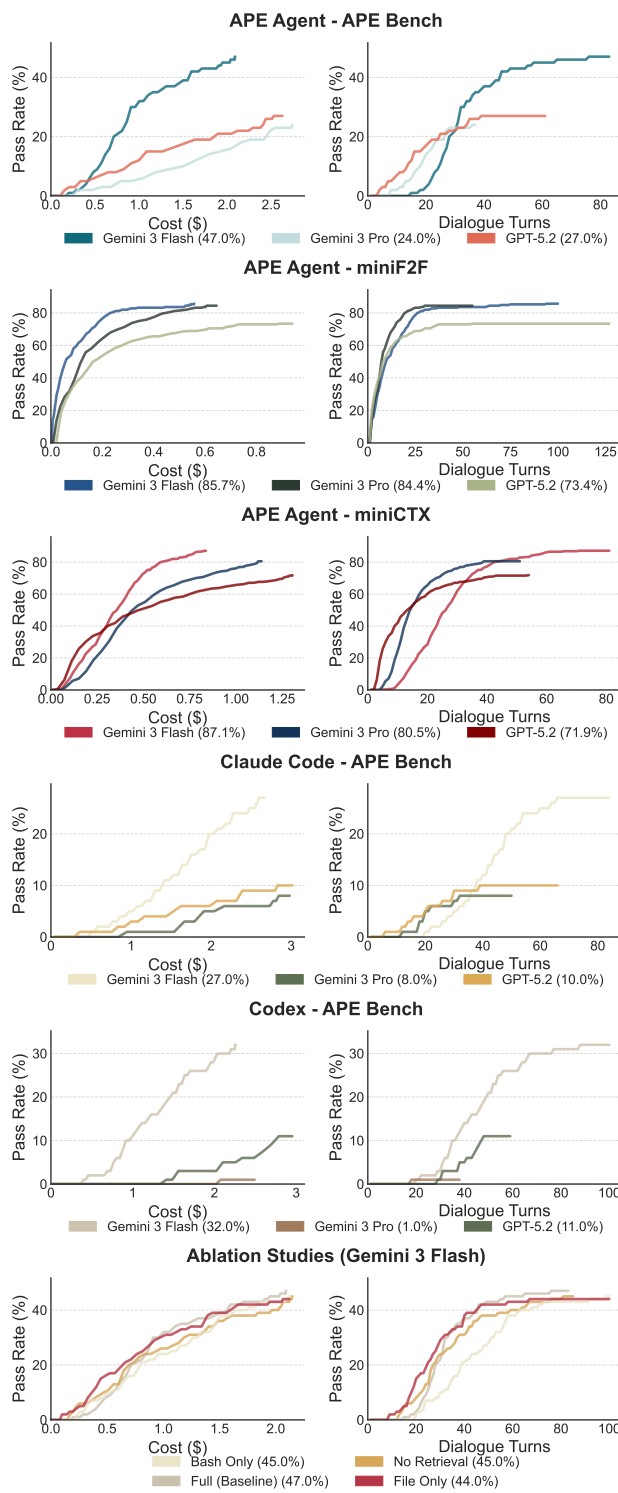

*Figure 3.* **Pass rate efficiency curves.** miniF2F saturates early while APE-Bench climbs without saturation, shifting evaluation from capability-limited to budget-limited. Notably, cost–turn comparison reveals Gemini 3 Flash's advantage is economic: model gaps compress per turn, with GPT-5.2 rising competitively before saturating at fewer turns.

## 6.2. Model Performance on APE-Bench

**Main results across benchmarks.** Under fixed $3 budgets per task, we track both cumulative cost and dialogue turns to disentangle cost-efficiency from reasoning capability. Figure 3 presents results across three benchmarks. On APE-Bench, Gemini 3 Flash achieves 47% success rate, substantially outperforming GPT-5.2 (27%) and Gemini 3 Pro (24%). On miniF2F, Gemini 3 Flash and Gemini 3 Pro achieve comparable performance (86%, 84%) while GPT-5.2 lags at 73%. On miniCTX, Gemini 3 Flash reaches highest performance (87.1%) followed by Gemini 3 Pro (80.5%) and GPT-5.2 (71.9%). Efficiency curves reveal stark difficulty differences: miniF2F exhibits rapid saturation with most models reaching 70-80% within $0.2, miniCTX shows intermediate efficiency requiring $0.4-0.6 to approach final performance, while APE-Bench demands gradual climbs through the full $3 budget. Table 4 shows miniCTX's mathlib tasks (64-90%) outperform APE-Bench (24-47%) by 1.9-3.4×, demonstrating repository-scale engineering imposes substantial difficulty beyond theorem proving.

| Model | miniCTX (mathlib) | miniCTX (others) | APE-Bench |
|---|---|---|---|
| Gemini 3 Flash | 90.0% | 86.6% | 47.0% |
| Gemini 3 Pro | 82.0% | 80.3% | 24.0% |
| GPT-5.2 | 64.0% | 73.2% | 27.0% |

*Table 4.* Success rates on miniCTX and APE-Bench by data source.

**Efficiency curves reveal the source of performance gaps.** Comparing cost and dialogue turn curves across all benchmarks reveals that Gemini 3 Flash's advantage stems from cost-efficiency rather than superior reasoning. The dialogue turn curves show models achieving comparable per-turn progress, with GPT-5.2 often demonstrating faster early saturation in challenging tasks (APE-Bench, miniCTX). However, higher per-turn costs cause premature budget exhaustion for Gemini 3 Pro and GPT-5.2, limiting total exploration opportunities. This explains why cost-based curves show larger performance gaps than turn-based curves: models demonstrate similar per-turn capabilities, but cost efficiency determines total attempts and final success rates under fixed budgets.

## 6.3. Task Characterization and Difficulty Analysis

Our 100 tasks span 17 mathematical domains and 67 unique Mathlib commits. We analyze task difficulty along multiple dimensions to characterize what makes repository-scale proof engineering challenging (full taxonomy in Appendix C).

**Modification pattern determines task structure.** Tasks decompose into five modification patterns (Table 5): *ad-*

| Pattern | N | Gemini 3 Flash | Gemini 3 Pro | GPT-5.2 |
|---|---|---|---|---|
| Additive Extension | 61 | 48% | 18% | 25% |
| Integration/API Update | 15 | 60% | 27% | 27% |
| Migration/Removal | 14 | 43% | 29% | 14% |
| New Module Creation | 9 | 33% | 33% | 56% |
| Modification Only | 1 | 0% | 100% | 100% |

*Table 5.* Pass rates by task modification pattern.

| Predictor | Bin | N | Gemini 3 Flash | Gemini 3 Pro | GPT-5.2 |
|---|---|---|---|---|---|
| Diff Lines ($r=-0.38$) | ≤15 | 27 | 74% | 33% | 26% |
| | 16–30 | 26 | 50% | 31% | 31% |
| | 31–60 | 19 | 47% | 21% | 32% |
| | 61–100 | 11 | 27% | 9% | 9% |
| | 100+ | 17 | 12% | 6% | 29% |
| Cross-File Sources ($r=-0.39$) | 0 | 5 | 80% | 60% | 80% |
| | 1–5 | 33 | 73% | 36% | 30% |
| | 6–15 | 43 | 37% | 19% | 16% |
| | 15+ | 19 | 16% | 0% | 32% |

*Table 6.* Pass rates by the two strongest difficulty predictors (Pearson $r$ with Gemini 3 Flash). Both show monotonic decline; GPT-5.2 shows anomalous resilience at high complexity.

*ditive extension* (61%—adding new declarations), *integration/API update* (15%), *migration/removal* (14%), *new module creation* (9%), and *modification-only* (1%). Per-pattern pass rates reveal that integration tasks are easiest (60% Gemini 3 Flash) while new module creation is hardest for Gemini 3 Flash (33%) but easiest for GPT-5.2 (56%)—suggesting module creation rewards stronger reasoning over iterative compilation.

**Modification size and cross-file complexity are the strongest difficulty predictors.** Table 6 shows pass rates binned by the two strongest difficulty predictors identified through correlation analysis (Appendix D). Diff size ($r=-0.38$) and cross-file source count ($r=-0.39$) both exhibit clear monotonic difficulty scaling for Gemini 3 Flash and Gemini 3 Pro: tasks with ≤15 diff lines achieve 74% (Gemini 3 Flash), dropping monotonically to 12% for 100+ lines; tasks with 0 cross-file sources pass at 80%, dropping to 16% for 15+ sources. Declaration count exhibits a similar pattern: tasks requiring 0 new declarations pass at 83%, while 10+ declarations drop to 25%. Notably, GPT-5.2 breaks the monotonic pattern at the highest complexity bins (29% at 100+ lines, 32% at 15+ cross-file sources), consistent with its reasoning-heavy, low-iteration strategy (§6.5) that amortizes high per-turn cost on tasks where fewer attempts suffice.

**Failure mode analysis reveals a hidden capability gap.** All failures across all models are `cost_limit`

| Scaffold | Model | Pass | Judge OK | Budget | Error |
|---|---|---|---|---|---|
| APE-Agent | Gem. 3 Flash | 47 | 5 | 48 | 0 |
| | Gem. 3 Pro | 24 | 5 | 71 | 0 |
| | GPT-5.2 | 27 | 36 | 37 | 0 |
| Claude Code | Gem. 3 Flash | 27 | 2 | 68 | 3 |
| | Gem. 3 Pro | 8 | 0 | 88 | 4 |
| | GPT-5.2 | 10 | 6 | 82 | 2 |
| Codex | Gem. 3 Flash | 32 | 3 | 51 | 14 |
| | Gem. 3 Pro | 1 | 0 | 28 | 71 |
| | GPT-5.2 | 11 | 8 | 80 | 1 |

*Table 7.* Failure mode decomposition. *Judge OK*: passed compilation and semantic verification but exceeded budget. GPT-5.2 with APE-Agent shows 36 hidden successes (potential 63% pass rate).

terminations—no turn-limit or crash failures occur. However, failure decomposition reveals a striking asymmetry (Table 7): among GPT-5.2's 73 failures, 36 (49%) had already produced solutions passing both compilation *and* semantic verification before exhausting the $3 budget. In contrast, 91% of Gemini 3 Flash failures and 93% of Gemini 3 Pro failures never reached successful compilation. This means GPT-5.2's *potential* pass rate with unlimited budget is 63% (27 passed + 36 hidden successes), far exceeding Gemini 3 Flash's 47%—strongly supporting our conclusion that Flash wins through cost-efficiency, not superior reasoning.

**Cross-file reasoning is essential.** Despite targeting single-file modifications, static analysis of gold solutions reveals that **95% of tasks (95/100) reference declarations defined in other Mathlib files** (median 9 unique cross-file declarations from 8 source files per task). This is not based on agent behavior (whether agents *choose* to retrieve), but on what the gold solution *requires*—a conservative lower bound, since our analysis only examines added lines and applies same-file-priority disambiguation. The 5 tasks with no cross-file dependencies are predominantly tactic metaprogramming tasks referencing Lean compiler APIs outside the Mathlib declaration index—their cross-file needs exist but are not captured by our Mathlib-focused analysis.

### 6.4. Semantic Judge Reliability

We validate our LLM-as-Judge pipeline through expert annotation of 64 agent solutions with three-dimensional ratings (semantic correctness, requirement alignment, scope control). All solutions passed syntactic verification (Lean compilation) before semantic evaluation. Semantic judges demonstrate consistent evaluation across all dimensions with 89-94% accuracy (Table 8), validating reliable semantic validation for proof engineering evaluation.

- *Syntactic verification enables reliable semantic evaluation.* Semantic judges achieve 92-94% accuracy on semantic correctness and requirement alignment. Syntactic veri-

| Dimension | Accuracy | FN | FP |
|---|---|---|---|
| Semantic Correctness (64) | 92.2% | 5 | 0 |
| Requirement Alignment (64) | 93.8% | 4 | 0 |
| Scope Control (64) | 89.1% | 3 | 4 |
| Overall (64) | 90.6% | 2 | 4 |

*Table 8.* Semantic judge accuracy by evaluation dimension.

| Scaffold (prompt) | Modify | Retrieve | Execute | Submit |
|---|---|---|---|---|
| APE-Agent (16.5K) | 37.8% | 52.4% | 5.6% | 4.1% |
| Claude Code (65K) | 22.8% | 56.4% | 16.9% | 1.2% |
| Codex (33K) | 7.1% | 60.3% | 26.6% | 1.0% |

*Table 9.* Tool usage (% of calls, averaged across models). Prompt size in parentheses. Modify = write + edit. APE-Agent's integrated verification reduces execute calls by 3–5×.

fication filters out solutions with obvious errors, enabling judges to focus on mathematical validity assessment.

- *Scope control shows reliable but improvable performance.* Scope control assessment achieves 89% accuracy, demonstrating reliable evaluation of engineering discipline— achieving objectives through focused, minimal changes. This slightly lower performance reflects the inherent complexity of assessing software engineering practices. Future work should refine scope assessment criteria.

### 6.5. Scaffold Interchangeability

We evaluate three scaffolds on identical tasks with identical services (Figure 3, row 4-5). APE-Agent outperforms Claude Code and Codex by 15-23 percentage points across models. Table 9 quantifies behavioral differences through tool usage distribution (averaged across models). Four implementation factors explain performance gaps:

- *Integrated verification eliminates redundant calls.* APE-Agent averages 5.6% execute calls versus 16.9% for Claude Code and 26.6% for Codex. APE-Agent binds file editing and compilation verification into single operations, while Claude Code and Codex require separate execute calls after each modification.

- *Verbose prompting increases baseline costs.* APE-Agent uses 16.5K characters for system prompt and tool definitions versus 33K for Codex and 65K for Claude Code. Longer prompts directly reduce available budget for task execution.

- *Conservative retrieval patterns consume additional budget.* Codex prompts encourage information-gathering behaviors before actions. Codex averages 60.3% retrieve calls versus 52.4% for APE-Agent, shifting budget from modification attempts to retrieval operations.

### 6.6. Tool Configuration

We ablate tool configurations on Gemini 3 Flash, varying file operations, shell execution, and semantic retrieval while retaining compilation verification (Table 10 and Figure 3, row 5). Four configurations produce nearly overlapping efficiency curves with final rates spanning only 44-47%. This pervasive stability indicates the primary bottleneck lies

| Configuration | File Ops | Shell | Retrieval | Pass Rate |
|---|---|---|---|---|
| Full | ✓ | ✓ | ✓ | 47% |
| No retrieval | ✓ | ✓ | ✗ | 45% |
| File only | ✓ | ✗ | ✗ | 44% |
| Bash only | ✗ | ✓ | ✗ | 45% |

*Table 10.* Tool configuration ablation results.

in reasoning capabilities rather than tool interfaces. Agents encounter the same capability ceiling regardless of tool availability, representing the current reasoning frontier for repository-scale proof engineering tasks.

### 6.7. Multi-Version Infrastructure

APE-Bench requires services across 67 library versions. Content-addressable storage enables efficient support through cryptographic hash-based deduplication. For execute services, deduplication reduces storage from 480GB to 71GB (85% reduction). For retrieve services, 15M declaration instances reduce to 229K unique declarations, cutting annotation cost from $12.9K to $197 (98.5% reduction). This efficiency enables practical multi-version evaluation and continual benchmark regeneration.

## 7. Conclusion

We introduced APE, extending formal mathematics evaluation from isolated theorem proving to repository-scale proof engineering through automated task extraction and contract-based evaluation infrastructure. Our evaluation reveals that under fixed budgets, cost-efficiency rather than reasoning capability determines model rankings, reframing the evaluation paradigm from capability-limited to budget-limited.

Current limitations include restriction to Lean/Mathlib with single-file modification tasks, and reliance on LLM-based semantic judges whose scope control—detecting unauthorized modifications beyond task specifications—remains the weakest evaluation dimension. Extending to additional formal ecosystems and multi-file engineering remains future work.

## Impact Statement

This paper presents evaluation infrastructure for formal mathematics. Improving AI capabilities for proof engineering could benefit mathematical formalization efforts and, in the longer term, verification of safety-critical systems. All benchmark data derives from publicly available Mathlib commits under the Apache 2.0 license. We release all code and infrastructure as open source.

## Acknowledgements

Xin and Li are supported by the AI for Math Fund. We thank the anonymous reviewers and area chairs for their helpful comments and feedback.

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

## A. Complete Task Example

This appendix provides a detailed walkthrough of a complete APE-Bench task to illustrate the full evaluation pipeline: task specification, gold solution, agent execution, and semantic validation.

### A.1. Task Specification

- Task ID: `pe_622f41ab_Sqrt`

- Repository: `mathlib4@622f41ab` (2026-01-12)

- Toolchain: `leanprover/lean4:v4.27.0-rc1`

- Target File: `Mathlib/Data/Real/Sqrt.lean` (477 lines, blocked)

- Instruction: Add limits at infinity for Real square root and automate continuity proof:

  1. Automate the proof of `continuous_sqrt` and register it with `fun_prop`.
  2. Prove that the image of the `atTop` filter under square root is `atTop`.
  3. Prove that the preimage (comap) of the `atTop` filter under square root is `atTop`.
  4. Establish that the square root function tends to infinity at infinity.

- Context: This task extends the real number square root API with filter-theoretic properties needed for analysis of limits at infinity. The task requires understanding Lean's filter library and proof automation tactics.

### A.2. Gold Solution

The human expert solution made the following changes (commit `622f41ab`, 13 lines modified):

```
@@ -122,8 +122,11 @@
-@[continuity]
-theorem continuous_sqrt : Continuous (sqrt· : R → R)
     := by
-   unfold sqrt
-   exact NNReal.continuous_coe.comp <|
-     NNReal.continuous_sqrt.comp
     continuous_real_toNNReal
+@[continuity, fun_prop]
+theorem continuous_sqrt : Continuous (sqrt· : R → R)
     := by unfold sqrt; fun_prop
+
+@[simp]
+lemma map_sqrt_atTop : map (sqrt·) atTop = atTop := by
+   unfold sqrt; change map (toReal ∘ sqrt ∘ toNNReal)
     atTop = atTop; simp [← map_map]
+
+@[simp] lemma comap_sqrt_atTop : comap (sqrt·) atTop =
     atTop := by [...]
+lemma tendsto_sqrt_atTop : Tendsto (sqrt·) atTop atTop
     := map_sqrt_atTop.le
```

**Key changes:**

- Automated continuity proof using `fun_prop` tactic

- Added three filter lemmas establishing sqrt behavior at infinity

- Used composition decomposition to leverage existing `NNReal` lemmas

### A.3. Agent Execution Trace

The agent (APE-Agent with Gemini 3 Flash) completed the task in 32 turns over 504 seconds (8.4 minutes), costing $0.79 with 1M tokens (615K cached).

**Phase 1: Exploration (Turns 1-7).** File reading and environment setup; identified `continuous_sqrt` theorem and sqrt definition.

**Phase 2: Implementation (Turns 8-25).** Implemented theorem scaffolds with library retrieval:

- Modified `continuous_sqrt` proof, added filter lemmas

- Used `lean_retrieve` to query 15+ Mathlib functions

- Retrieved: `tendsto_atTop_atTop`, `NNReal.sqrt.map_atTop`, `comap_atTop_eq_of_gc`

- Agent reasoning (Turn 25): "Critical dependency issue: sqrt is irreducible. Need to decompose as `Real.toNNReal ∘ NNReal.sqrt ∘ NNReal.toReal` composition."

**Phase 3: Critical Fix (Turn 26).** KEY INSIGHT: `rw [sqrt]` fails on irreducible definition. Solution: Changed to `unfold sqrt`.

**Phase 4: Verification (Turns 27-32).** Polished proofs, verified all 4 requirements, submitted result with score=1.0.

**Key capabilities demonstrated:**

- Systematic library knowledge retrieval through `lean_retrieve`

- Iterative debugging (Turn 26 irreducibility insight)

- Strategic composition decomposition to leverage existing lemmas

### A.4. Evaluation Results

**Compilation verification.** PASS — All modifications type-check against `leanprover/lean4:v4.27.0-rc1`, Mathlib at `622f41ab`.

**Semantic validation.** LLM Judge Assessment (3 judges, majority vote):

- Judge 1: Accept — Semantic: Excellent, Requirement: Excellent, Scope: Good

  Reasoning: "Proofs use correct library lemmas. All requirements met. Minor cosmetic edits acceptable."

- Judge 2: Accept — Semantic: Excellent, Requirement: Excellent, Scope: Excellent

  Reasoning: "Perfect implementation following library conventions. Proofs are direct and use standard techniques."

- Judge 3: Accept — Semantic: Excellent, Requirement: Excellent, Scope: Good

  Reasoning: "Mathematically correct. Composition-based proofs for filter lemmas are exactly right."

Majority vote: Accept (3/3)

Human Expert Evaluation: Pass — Semantic: Excellent, Requirement: Excellent, Scope: Excellent

"Completely correct proofs. All four objectives implemented. Only modified necessary code."

**Agreement.** Both LLM judge (3/3 accept) and human expert (Pass) agree on acceptance, representing the typical case (90.6% agreement, Table 8).

# B. Semantic Validation Case Studies

We analyze representative disagreement cases between LLM judge and human evaluation, focusing on the failure modes identified in §6.4.

### B.1. False Positive: Unauthorized Deletion

**Task.** `pe_c1d772d6_Basic` — Establish naturality of iterated slice equivalence and connection to post functor in `CategoryTheory/Comma/Over/Basic.lean` (1220 lines).

**Agent solution.** Correctly implemented all three required definitions (`iteratedSliceForwardNaturalityIso`, `iteratedSliceEquivOverMapIso`, `iteratedSliceForwardIsoPost`). However, the agent also deleted two existing declarations not mentioned in the task:

- Deleted `essImage.of_overPost`

- Deleted `essImage.of_underPost`

**LLM Judge.** 3/3 Accept

Common reasoning: "Each new theorem matches the task description. All requirements implemented."

Missed: Unauthorized deletion of existing declarations.

**Human Expert.** Fail — Semantic: Excellent, Requirement: Excellent, Scope: Unacceptable.

"Task did not ask to delete `essImage.of_overPost` or `essImage.of_underPost`."

**Analysis.** Scope detection failure. LLM judges verified that required additions were correct but did not check whether existing code was preserved. Deletion of unrelated declarations breaks downstream dependencies. Improvement direction: Explicit deletion detection in judge prompts.

### B.2. False Positive: Excessive Scope

**Task.** `pe_f56df434_Multiplication` — Formalize torsion-free property for Hahn modules (add one instance).

**Scope violations beyond core objective:**

- Deleted 15 existing lemmas (lines 120-180)

- Rewrote 8 proofs with different tactics (no behavioral change)

- Modified 6 lemma names for "consistency" (breaks downstream dependencies)

- Changed import structure

**LLM Judge.** 3/3 Accept — "Core instance correct. Refactoring improves clarity."

**Human Expert.** Fail — "Required instance is perfect, but massive out-of-scope refactoring due to poor scope control."

**Analysis.** Judges did not penalize scope violations when core mathematical content was correct. Improvement direction: Explicit scope metrics (diff size bounds, deletion warnings, style consistency checks).

### B.3. False Negative: Calibration Inconsistency

**Task.** `pe_5136020e_IsTensorProduct` — Prove pushout property for tensor products.

**Agent solution.** Correctly implemented required theorem. Minor benign refactorings:

- Made 4 implicit arguments explicit for clarity (lines 211-215)

- Removed redundant type annotations (5 locations)

- Added explicit instance declarations (3 locations)

All changes improve code clarity without affecting behavior.

**LLM Judge.** 1/3 Accept (Rejected)

- Judge 1: Reject — "Modified unrelated code"

- Judge 2: Reject — "Scope exceeds requirements"

- Judge 3: Accept — "Minor improvements acceptable"

**Human Expert.** Pass — "Changes improve readability. Acceptable minor refactoring despite small scope deviations."

**Analysis.** Inter-judge calibration variance. Two judges applied strict interpretation ("only modify exactly what's stated"), one allowed beneficial refactoring. Improvement direction: Explicit scope tolerance guidelines or reference-based calibration examples.

## C. Task Taxonomy and Dataset Statistics

This section provides a comprehensive characterization of the 100 APE-Bench tasks along multiple dimensions, complementing the main text analysis in §6.3.

### C.1. Dataset Summary

Table 11 summarizes continuous metrics across all tasks.

| Metric | Min | Median | Mean | Max |
|---|---|---|---|---|
| Diff lines | 10 | 30 | 52 | 284 |
| Gold diff lines | 21 | 62 | 99 | 553 |
| Original file lines | 35 | 410 | 489 | 1345 |
| Instruction length (chars) | 114 | 484 | 470 | 838 |
| Instruction words | 23 | 65 | 67 | 121 |
| Added declarations | 0 | 3 | 5.3 | 36 |
| Total declaration changes | 1 | 5 | 7.8 | 50 |
| Touched declarations | 1 | 5 | 7.0 | 46 |
| Unique commits | | 67 | | |
| Math domains | | 17 | | |
| Date range | | 2026-01-06 to 2026-01-12 | | |

*Table 11.* APE-Bench dataset summary statistics. All tasks require $\geq 10$ diff lines (median 30), with median 5 declaration changes per task.

## D. Difficulty Predictor Analysis

Figure 4 shows the Pearson correlation between task properties and success for each model.

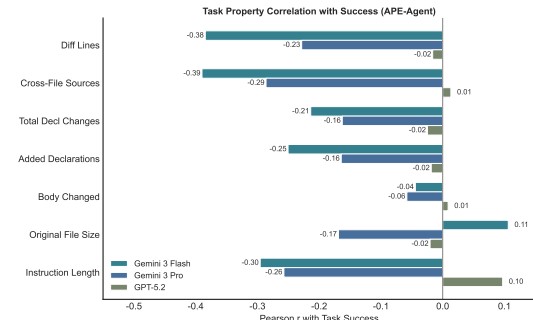

*Figure 4.* **Task property correlation with success.** Cross-File Sources ($r=-0.39$ for Gemini 3 Flash) and Diff Lines ($r=-0.38$) are the strongest difficulty predictors. All properties show consistent negative correlations across models.

Key correlations with success:

- **Cross-file complexity and modification scale dominate:** Cross-File Sources ($r=-0.39$) and Diff Lines ($r=-0.38$) are the two strongest predictors for Gemini 3 Flash, followed by Total Declaration Changes ($r\approx-0.21$). For Gemini 3 Pro, Cross-File Sources ($r=-0.29$) also exceeds Diff Lines ($r=-0.23$).

- **Consistency across models:** All three models show the same ordering of difficulty predictors, suggesting these task properties reflect intrinsic complexity rather than model-specific weaknesses.

- **File size is not predictive:** Original file size shows near-zero correlation ($r\approx0$), indicating that the amount of code to *change* matters more than the amount of code to *read*.

The causal chain is: complex tasks (many cross-file dependencies, high diff lines) → more agent effort (retrieval, file reading, compilation attempts) → budget exhaustion → failure. This confirms that the primary bottleneck is reasoning capability under fixed budgets, not tool availability.

