# OpenReview forum: "APE-Bench: Evaluating Automated Proof Engineering for Formal Math Libraries"
_ICML.cc/2026/Conference — ICML 2026 regular_

### Official Review · Reviewer_x4Q8 · 2026-03-10

**Soundness:** 2
**Presentation:** 2
**Significance:** 2
**Originality:** 2
**Overall Recommendation:** 4
**Confidence:** 3

**Summary:**

This paper introduces APE, a framework designed to evaluate automated proof engineering on formal mathematics repositories. It presents APE-Bench, a benchmark derived from the commit history of Mathlib, and APE-Harness, an infrastructure for executing and verifying repository-scale proof engineering tasks. The proposed benchmark is used to evaluate the performance of several large language models (LLMs).

**Compliance With Llm Reviewing Policy:**

Affirmed.

**Final Justification:**

The author clarify why a new benchmark is needed and I think it is neccessary to propose this benchmark.

**Key Questions For Authors:**

1. What mechanisms are used to verify the correctness and quality of the automatically generated instructions from commit histories? Is there any human verification or filtering beyond LLM-based validation?

2. Can the authors provide a more detailed analysis of the benchmark itself (e.g., task difficulty distribution, task categories, or comparison with existing datasets) to better justify its usefulness?

**Limitations:**

yes

**Strengths And Weaknesses:**

**Strenghs**

1. The paper studies the evaluation of LLM agents in formal mathematics development beyond isolated theorem proving.

2. The idea of constructing tasks from real repository commits is potentially interesting.

3. The paper provide an execution infrastructure for running such tasks.

---

**Weaknesses**

1. The paper appears rushed and contains many formatting problems. The manuscript contains multiple formatting issues (e.g., text overlapping in tables, content exceeding page margins, and inconsistent layout), which significantly reduce readability and suggest the paper was not carefully prepared.

2. The paper claims that existing benchmarks are insufficient; however, it does not clearly justify the necessity of introducing a new benchmark. Beyond MiniF2F, several benchmarks have already been proposed that go beyond simple theorem-proving tasks. For example, [1] formulates problem synthesis as a step-by-step agentic exploration process rather than a one-shot generation approach.

3. The generated data lacks strong validation. The benchmark relies heavily on automatically generated instructions on commit histories and LLM-based validation. The paper provides limited evidence that the generated tasks are correct, meaningful, or representative.

4. The evaluation analysis is limited. The experiments mainly report success rates, but lack deeper analysis of task difficulty, benchmark characteristics, or insights into model behavior, which are typically provided in prior benchmark studies.

[1] Qi et al. "Let's Explore Step by Step: Generating Provable Formal Statements with Deductive Exploration"

---

> ### Author Rebuttal · Authors · 2026-03-30
>
> We thank Reviewer x4Q8 for the detailed review.
>
> ### W1: Formatting problems
>
> All formatting issues (text overlaps, table overflows, margin violations) will be fixed in the camera-ready version.
>
> ### W2: Not clearly justified why a new benchmark is needed
>
> We clarify the distinction. [Qi et al.] takes existing theories as input and generates new theorem *statements* through deductive exploration, evaluated by whether generated statements are provable. APE-Bench takes a repository and natural-language engineering instructions as input, and requires *code modifications* to an existing codebase, evaluated by compilation + semantic judgment of requirement satisfaction. The two differ in input (theory vs. repository+instruction), output (theorem statements vs. code edits), and evaluation criteria (provability vs. engineering correctness). This parallels how SWE-Bench complements LeetCode in software engineering—both involve code, but the skills tested are fundamentally different. miniF2F, FATE, and miniCTX similarly focus on *proving given statements*, not on engineering modifications.
>
> **Empirical evidence strongly supports this distinction.** The same models achieve 64–90% on miniF2F/miniCTX but only 24–47% on APE-Bench—a **1.9–3.4× performance gap** (Table 1 in paper). We evaluate on miniF2F and miniCTX through the same infrastructure, demonstrating complementarity rather than replacement.
>
> ### W3: Generated data lacks strong validation / Q1: Validation of automatically generated instructions?
>
> Our pipeline provides multi-stage validation:
>
> 1. **Commit filtering** ensures substantive, coherent changes (5–100 diff lines filtering threshold, mathematical content, type-checking).
> 2. **Instruction synthesis** uses agents with full repository context, generating intent-based instructions rather than diff descriptions.
> 3. **Automated validation** verifies that the original commit patch satisfies the generated instruction under dual verification (compilation + semantic judgment). If the generated instruction were incorrect, the gold commit would fail verification.
> 4. **Source credibility**: all tasks originate from **merged Mathlib commits** that passed Mathlib's CI and maintainer review.
>
> Instructions are not individually human-verified, but the above consistency checking against human-reviewed gold solutions provides a strong correctness guarantee.
>
> ### W4: Evaluation analysis is limited / Q2: More detailed analysis?
>
> **Task characterization:** Tasks span 5 modification patterns, 17 math domains, and 10–284 non-blank added lines (median 30). Pass rates decrease monotonically with modification size—from 74% (≤15 lines) to 12% (100+ lines) for Gemini 3 Flash. Tasks modifying only proof bodies achieve 90% pass rate vs. 42% for tasks involving declaration headers.
>
> **Cross-file dependency analysis (static, gold-solution-based):** 95/100 tasks (95%) reference cross-file Mathlib declarations (median 9 declarations from 8 source files). Cross-file source count is the single strongest difficulty predictor (r=-0.39):
>
> | Cross-File Sources | N | Gemini 3 Flash | Gemini 3 Pro | GPT-5.2 |
> |---|---|---|---|---|
> | 0 | 5 | 4/5 (80%) | 3/5 (60%) | 4/5 (80%) |
> | 1–5 | 33 | 24/33 (73%) | 12/33 (36%) | 10/33 (30%) |
> | 6–15 | 43 | 16/43 (37%) | 8/43 (19%) | 7/43 (16%) |
> | 15+ | 19 | 3/19 (16%) | 0/19 (0%) | 6/19 (32%) |
>
> Cross-model difficulty tiers reveal 41 unsolved tasks (0/3 models solve) concentrating in abstract algebra domains, with ~2.5× larger median diff size than easy tasks. Full difficulty predictor correlations and per-domain breakdowns will be included in the camera-ready version.

---

> > ### Author Rebuttal · Reviewer_x4Q8 · 2026-04-04
> >
> > Thanks for your detailed rebuttals and I will raise my score to 4.

---

### Official Review · Reviewer_SYi2 · 2026-03-12

**Soundness:** 3
**Presentation:** 3
**Significance:** 3
**Originality:** 3
**Overall Recommendation:** 5
**Confidence:** 3

**Summary:**

This paper considers the problem of repository-level proof engineering for formal mathematics. In contrast to prior benchmarks that primarily evaluate isolated theorem-proving tasks, the authors propose APE-Bench, which targets more realistic proof-engineering scenarios . To support this evaluation, the paper further introduces APE-Harness as an execution framework. The authors then conduct extensive experiments on this benchmark to compare different agentic systems and tool configurations.

**Compliance With Llm Reviewing Policy:**

Affirmed.

**Final Justification:**

I will keep my original score 5 (accept).

**Key Questions For Authors:**

This paper argues that reasoning, rather than tool access, is the main bottleneck. Could this conclusion depend on the current scaffolds and their ability to exploit the available tools?

**Limitations:**

Yes

**Strengths And Weaknesses:**

Strengths:

Significance:
This paper targets an important problem. Research in formal mathematics has so far focused largely on isolated theorem-proving tasks, but the field is increasingly moving toward more realistic, repository-scale settings that require proof engineering over large formal libraries. This paper addresses a meaningful gap by introducing a benchmark specifically designed for repo-level formal math tasks.

Originality:
The paper is reasonably novel in its problem formulation. Rather than proposing yet another benchmark for isolated proof synthesis, it introduces Automated Proof Engineering (APE) as a distinct evaluation setting for repository-scale formal mathematics.

Soundness:
The paper is reasonably solid from an empirical and systems perspective. It provides experimental results on the proposed benchmark and evaluates multiple models and configurations. In addition, the introduction of APE-Harness strengthens the work by providing a unified execution framework, which makes the evaluation pipeline clearer and improves the reproducibility and practicality of the benchmark.

Presentation:
The paper is generally well presented. The motivation is clear, and the distinction between traditional theorem proving and repository-scale proof engineering is explained in a fairly intuitive way. The overall structure is also easy to follow.

---

> ### Author Rebuttal · Authors · 2026-03-30
>
> We thank Reviewer SYi2 for the positive evaluation and the insightful question.
>
> ### Q1: Could the "reasoning bottleneck" conclusion depend on current scaffolds?
>
> This is a valid observation. Our evidence comes from multiple complementary sources:
>
> 1. **Three diverse scaffolds** with fundamentally different architectures (APE-Agent: ReAct with integrated verification; Claude Code: progress-tracking pattern; Codex: execution-encapsulation pattern) all hit similar ceilings.
>
> 2. **Four tool ablation configurations** span only 44–47% pass rate. The cross-success matrix shows pervasive overlap—33–35 tasks solved by every pair, with only 1–3 unique to any single configuration:
>
> | | Full (47) | No Retrieve (45) | Bash Only (45) | File Only (44) |
> |---|---|---|---|---|
> | Full | 47 | 35 | 35 | 34 |
> | No Retrieve | 35 | 45 | 34 | 33 |
> | Bash Only | 35 | 34 | 45 | 34 |
> | File Only | 34 | 33 | 34 | 44 |
> | *Unique* | 3 | 3 | 1 | 2 |
>
> 3. **Agent behavior analysis** shows failed tasks use 3× more tool calls than successful tasks (median 1431 vs. 496 for Gemini 3 Flash). Three distinct strategies emerge: Gemini 3 Flash uses high-iteration (31 turns, 117 compilation attempts, lowest per-turn cost); GPT-5.2 uses low-iteration (16 turns, near-zero compilation attempts, highest reasoning per turn); Gemini 3 Pro takes an intermediate approach. Despite these different strategies, all hit similar ceilings.
>
> 4. **Cross-file complexity as the dominant predictor** (r=-0.39) further supports that the bottleneck is *understanding* cross-file dependencies, not *accessing* them—all configurations provide access, but reasoning about the retrieved information is the limiting factor.
>
> However, we agree this is a **snapshot conclusion**: future scaffolds incorporating more sophisticated planning (e.g., tree search over proof strategies, learned tool-use policies, or multi-agent collaboration) could potentially shift the bottleneck.

---

> > ### Author Rebuttal · Reviewer_SYi2 · 2026-04-05
> >
> > The authors' response addressed my question.

---

### Official Review · Reviewer_u6Rz · 2026-03-13

**Soundness:** 3
**Presentation:** 1
**Significance:** 3
**Originality:** 3
**Overall Recommendation:** 5
**Confidence:** 3

**Summary:**

This paper introduces Automated Proof Engineering (APE) as a more realistic evaluation setting for formal-math agents. Instead of focusing only on isolated theorem proving, the paper studies formal proving and other relevant tasks under repository-level settings with controlled environment. The authors propose APE-Bench, a benchmark mined from real Mathlib commits, and APE-Harness, a task-contract-based framework for standardized execution and evaluation across agent scaffolds, and APE-Agent, a ReAct-style white-box agent baseline. The evaluation shows that these proof-engineering tasks are substantially harder than traditional theorem-proving benchmarks, and the paper also includes judge validation, scaffold comparison, and infrastructure analysis to support the proposed benchmark and framework.

**Compliance With Llm Reviewing Policy:**

Affirmed.

**Final Justification:**

The rebuttal addressed all my questions and I keep the "accept" score.

**Key Questions For Authors:**

1/ How many benchmark tasks actually require cross-file reasoning, and how many are essentially single-file edits with repository context? A quantitative breakdown would help calibrate the “repository-scale” framing.

2/ Do the authors expect the benchmark construction pipeline and APE-Harness abstraction to generalize beyond Lean and Mathlib? What are the potential difficulties and how generalizable the APE-Harness is?

**Limitations:**

No, this paper could benefits from an explicity limitation discussion, including the LLM-based semantic judging and the Lean/Mathlib-only scope.

**Strengths And Weaknesses:**

**Strengths**

1/ The paper addresses a real and important gap in formal-math evaluation. Instead of only testing isolated theorem proving, it studies instruction-driven proof engineering in pinned repository environments, which is closer to realistic library maintenance and agent workflows. I also appreciate the use of recent real commits and continual benchmark regeneration, which helps reduce data contamination concerns and makes the benchmark more timely.

2/ The benchmark and infrastructure contributions are meaningful. APE-Bench combines natural-language instructions, repository/toolchain pinning, blocked target files, and dual verification, while APE-Harness provides a clean task-contract abstraction that makes evaluation more standardized and reproducible.

3/ The experimental study is fairly thorough. The paper does not only report benchmark performance, but also includes scaffold comparison, tool ablations, judge validation, and infrastructure analysis, which together strengthen the paper.

4/ The presentation content is clear and easy to follow. The problem setting, benchmark construction, and evaluation are all explained in a structured way.

**Weaknesses**

1/ Presentation: There are multiple text overlaps and table overflows. Please fix them.

2/ Limited scope of evaluation ecosystem: the benchmark is currently instantiated only in Lean and Mathlib. This is a reasonable and important domain, but it does narrow the evidence for broader generality. At minimum, I think the paper should discuss more explicitly whether the proposed benchmark construction and harness ideas are expected to transfer to other formal-math or verification ecosystems, including other projects to Lean and/or other formal frameworks like Coq/Dafny/Isabelle.

---

> ### Author Rebuttal · Authors · 2026-03-30
>
> We thank Reviewer u6Rz for the positive assessment and thoughtful suggestions.
>
> ### W1: Formatting issues
>
> We have identified all text overlaps, table overflows, and layout inconsistencies and will fix them in the camera-ready version.
>
> ### W2: Limited scope — Lean/Mathlib only
>
> The pipeline is parametric over any Git-hosted formal project, and the task contract abstraction (environment bindings, objectives, boundaries, verification protocols) is language-agnostic. Applying to Coq requires substituting the compilation backend; analogous library ecosystems exist (Mathcomp for Coq, AFP for Isabelle, verified projects in Dafny/F*). We chose Lean/Mathlib because it is the most actively developed community library (thousands of monthly commits), providing the richest source of engineering tasks. We will add explicit generalizability discussion and a Limitations paragraph in the camera-ready version.
>
> ### Q1: How many tasks require cross-file reasoning vs. single-file edits?
>
> This is the key question we address with rigorous evidence. Rather than citing agent behavior (whether agents *choose* to call retrieval tools), we performed **static analysis of gold solutions** to determine what cross-file knowledge each task *objectively requires*.
>
> **Method.** For each task, we extract identifiers from the gold diff's added lines, resolve them against the Mathlib commit index, and check whether the resolved declaration's source file differs from the target file. We apply conservative disambiguation: ambiguous matches default to same-file; only added lines (not context) are analyzed; unqualified short names matching >5 files are excluded. Every design choice *underestimates* cross-file dependency, making our numbers a lower bound.
>
> **Results:**
>
> | Metric | Value |
> |---|---|
> | Tasks requiring cross-file knowledge | **95/100 (95%)** |
> | Cross-file unique declarations / task (median) | **9** |
> | Cross-file source files / task (median) | **8** |
> | Cross-file source files / task (mean) | 10.6 |
>
> **Cross-file complexity is the single strongest difficulty predictor** (r=-0.39 for Gemini 3 Flash, exceeding diff lines at r=-0.38). Pass rate drops from 80% (0 cross-file sources) to 16% (15+ sources).
>
> The 5 tasks with zero cross-file dependencies are predominantly tactic metaprogramming tasks (`Tactic/Translate/Core.lean`) that reference Lean compiler APIs outside the Mathlib declaration index—their cross-file needs exist but are not captured by our Mathlib-focused analysis.
>
> While each modification targets a single file, the distinction between "single-file edit" and "no cross-file reasoning" is critical—repository context navigation across dozens of source files is essential for virtually every task.
>
> ### Q2: Generalizability beyond Lean/Mathlib?
>
> The pipeline design is parametric over repository, commit range, compilation backend, and retrieval service (semantic search against the target version's declaration index). The engineering components—compilation verification, version pinning, declaration retrieval—are standard for all major proof assistants and represent straightforward adaptation work.
>
> The more fundamental factor for generalization is **commit history quality**. Mathlib has the highest-quality commit history among formal libraries: strict CI, mandatory maintainer review, and consistent engineering conventions. This ensures that extracted tasks are well-defined and gold solutions are reliable. Other formal libraries (Mathcomp for Coq, AFP for Isabelle) vary in commit discipline, which would affect the quality and quantity of extractable tasks. The pipeline itself generalizes readily; the quality of the resulting benchmark depends on the source library's development practices.

---

> > ### Author Rebuttal · Reviewer_u6Rz · 2026-04-03
> >
> > The rebuttal addressed all my questions.

---

### Official Review · Reviewer_zCeC · 2026-03-13

**Soundness:** 2
**Presentation:** 4
**Significance:** 2
**Originality:** 3
**Overall Recommendation:** 4
**Confidence:** 3

**Summary:**

This paper build a benchmark that tests if AI can do proof maintenance work in Mathlib, not just prove one theorem, but edit files, add lemmas, fix proofs, like a real developer would. They pull 100 tasks from actual Mathlib commits, give models plain English instructions, and check if the result compiles and does what was asked. Best model (Gemini Flash) solves 47%.

**Compliance With Llm Reviewing Policy:**

Affirmed.

**Final Justification:**

The rebuttal addressed and resolved my concerns, hence raise score.

**Key Questions For Authors:**

Do agents have internet access during evaluation or how did you forbidden that?
Can you show pass rates by task type (refactoring vs adding lemmas vs proof automation)?

**Limitations:**

Small benchmark (100 tasks)
Possible runtime data leakage through internet access not addressed.
The paper's heavy abstraction language makes me (personally) harder to evaluate than it should be.

**Strengths And Weaknesses:**

Strengths.
Good idea — real Lean work is editing code, not just proving isolated statements, and nobody tested that before. Tasks come from real commits, not made up. Checking both "does it compile" and "did you do what I asked" makes sense. The finding that Gemini-Flash wins because it's cheap (more tries), not because it's smarter, is a useful insight. Tool ablation is clean — swapping tools barely changes scores, so the bottleneck is thinking, not tooling. Comparing three scaffolds on the same tasks is fair.

Weaknesses.
No breakdown of what kinds of tasks models pass or fail. (If the commit is just simple task or edited 1 or 2 line of script will bias the success rate)
100 tasks only for a benchmark.
The AI judge gets scope control wrong 23% of the time, and scope control is the whole point of testing engineering vs just proving.
The appendix shows the judge accepting a solution that deleted 15 existing lemmas it shouldn't have. If the judge misses stuff this obvious, the scores are hard to trust

---

> ### Author Rebuttal · Authors · 2026-03-30
>
> We thank Reviewer zCeC for the recognition that "real Lean work is editing code, not just proving isolated statements, and nobody tested that before."
>
> ### C1: No breakdown of what kinds of tasks models pass or fail; possible bias from simple 1–2 line edits
>
> **No trivial tasks.** All 100 tasks require >=10 non-blank added lines (median 30, range 10–284), with median 5 declaration changes per task. No task involves a simple 1–2 line edit—our commit filtering explicitly requires substantive mathematical content changes. Dataset spans 67 unique commits and 17 math domains.
>
> **Pass rate analysis.** Pass rates decrease monotonically with modification size—from 74% (≤15 lines) to 12% (100+ lines) for Gemini 3 Flash. Tasks modifying only proof bodies achieve 90% vs. 42% for tasks involving declaration headers.
>
> *Cross-file complexity* is the single strongest difficulty predictor (r=-0.39 for Gemini 3 Flash). For each task, we resolve identifiers in the gold diff's added lines against the Mathlib commit index and count unique cross-file source files:
>
> | Cross-File Sources | N | Gemini 3 Flash | Gemini 3 Pro | GPT-5.2 |
> |---|---|---|---|---|
> | 0 | 5 | 4/5 (80%) | 3/5 (60%) | 4/5 (80%) |
> | 1–5 | 33 | 24/33 (73%) | 12/33 (36%) | 10/33 (30%) |
> | 6–15 | 43 | 16/43 (37%) | 8/43 (19%) | 7/43 (16%) |
> | 15+ | 19 | 3/19 (16%) | 0/19 (0%) | 6/19 (32%) |
>
> *Task modification patterns:* Tasks decompose into five patterns—additive extension (61), integration/API update (15), migration/removal (14), new module creation (9), modification-only (1). Integration/API updates are easiest (60% Gemini 3 Flash), while new module creation is hardest for Gemini 3 Flash (33%) but easiest for GPT-5.2 (56%).
>
> *Math domains:* AlgebraicGeometry is the hardest (0–12% across models), while Topology is relatively easier (67% Gemini 3 Flash). Unsolved tasks (41/100, 0/3 models) concentrate in abstract algebra domains with ~2.5× larger median diff size than easy tasks.
>
> Due to rebuttal space constraints, we present selected highlights here; full per-domain pass rates, difficulty predictor correlations, and dataset summary statistics will be included in the camera-ready version.
>
> ### C2: 100 tasks only for a benchmark
>
> We agree 100 is a modest instantiation. Each task is substantially more complex than entries in miniF2F (244 problems) or miniCTX (334 theorems), requiring 10–284 non-blank added lines across median 5 declarations, with 95% requiring cross-file knowledge from median 8 source files. Our pipeline supports continual regeneration—the current instantiation deliberately uses a one-week window (2026-01-06 to 2026-01-12) for contamination control, not due to pipeline limitations.
>
> ### C3: Scope control accuracy (23% error) undermines trust
>
> We appreciate this concern. After submission, we discovered a **display truncation bug in our annotation interface** that caused annotators to assess scope control with incomplete modification context, disproportionately inflating false negatives. After fixing the tool and re-annotating all 64 solutions:
>
> | Dimension | Submitted | Revised |
> |---|---|---|
> | Semantic Correctness | 95.2% (2 FN, 1 FP) | 92.2% (5 FN, 0 FP) |
> | Requirement Alignment | 95.1% (2 FN, 1 FP) | 93.8% (4 FN, 0 FP) |
> | **Scope Control** | **77.0% (9 FN, 5 FP)** | **89.1% (3 FN, 4 FP)** |
> | Overall | 87.5% (2 FN, 6 FP) | 90.6% (2 FN, 4 FP) |
>
> This was a measurement artifact in the annotation tool, not a judge pipeline issue. The other two dimensions show minor shifts because each solution is judged on all three dimensions simultaneously—with complete context, some disagreements were reclassified across dimensions.
>
> The case referenced by the reviewer (§B.1, deleted 15 lemmas) was directly affected: after re-annotation with complete context, the human expert confirmed the judge's acceptance was correct (rated excellent). All benchmark tasks, model evaluation results, and pass rates are completely unchanged.
>
> ### Q1: Do agents have internet access?
>
> **No.** Agents interact exclusively with APE-Harness infrastructure services (file operations, compilation, declaration search) within isolated workspaces with OS-level permission enforcement. We will add an explicit statement to §6.1 in the camera-ready version.
>
> ### Q2: Pass rates by task type?
>
> See C1 above. By modification pattern:
>
> | Pattern | N | Gemini 3 Flash | Gemini 3 Pro | GPT-5.2 |
> |---|---|---|---|---|
> | Additive Extension | 61 | 29/61 (48%) | 11/61 (18%) | 15/61 (25%) |
> | Integration/API Update | 15 | 9/15 (60%) | 4/15 (27%) | 4/15 (27%) |
> | Migration/Removal | 14 | 6/14 (43%) | 4/14 (29%) | 2/14 (14%) |
> | New Module Creation | 9 | 3/9 (33%) | 3/9 (33%) | 5/9 (56%) |

---

> > ### Author Rebuttal · Reviewer_zCeC · 2026-04-02
> >
> > The rebuttal resolved my question.

---

### Decision · Program_Chairs · 2026-04-30

**Decision:**

Accept (regular)

**Comment:**

This paper introduces a timely and valuable framework for evaluating formal mathematics agents at the repository scale, successfully shifting the paradigm from isolated theorem proving to realistic, multi-file proof engineering. The reviewers unanimously supported acceptance, noting that the authors' excellent rebuttal fully resolved initial concerns regarding task difficulty, judge accuracy, and benchmark necessity with compelling quantitative evidence. I recommend acceptance for its strong technical contribution, with the strict condition that the authors meticulously fix the severe formatting issues present in the initial submission and incorporate their insightful rebuttal statistics into the camera-ready version.